# Dairy Cows Grazing Plantain-Based Pastures Have Increased Urine Patches and Reduced Urine N Concentration That Potentially Decreases N Leaching from a Pastoral System

**DOI:** 10.3390/ani13030528

**Published:** 2023-02-02

**Authors:** Thi Truong Nguyen, Soledad Navarrete, David Horne, Daniel Donaghy, Racheal H. Bryant, Peter Kemp

**Affiliations:** 1School of Agriculture and Environment, Massey University, Palmerston North 4410, New Zealand; 2Campus in Kon Tum, The University of Danang, 704 Phan Dinh Phung, Kon Tum 580000, Vietnam; 3Faculty of Agriculture and Life Sciences, Lincoln University, Lincoln 7647, New Zealand

**Keywords:** *Plantago lanceolata*, nitrogen excretion, urination frequency, urination behaviour, urine volume

## Abstract

**Simple Summary:**

Incorporating forage plantain into the diet of grazing dairy cows has the potential to reduce nitrogen (N) losses from pastoral systems. One of the key mechanisms for reduced N loss from plantain-fed cows is through increased urine volume and diluted urine N concentration, which increases the % of urine N used for plant growth. The aim of this study was to determine whether lower proportions of plantain in the diet would be effective in achieving changes in urination. The results showed that compared to cows grazing traditional perennial ryegrass and white clover pastures, cows with 25% plantain in their apparent dietary intake increased urine volume and urination frequency and reduced urine N% in the morning. When dietary plantain was less than 20%, there was only a small effect on increased urine volume, though morning urine N% was reduced. Ensuring more than 20% plantain in the diet has the potential to reduce N leaching risk via reduced urinary N load from grazing dairy cows.

**Abstract:**

The objective of this study was to determine the effect of grazing plantain-based pastures on urine volume, urination frequency, and urinary nitrogen (UN) concentration of dairy cows under a typical pastoral dairy practice offering approximately 25% supplemented feeds. The experiment was a completely randomised design with three pasture treatments (perennial ryegrass–white clover (RGWC); RGWC + low plantain rate (PLL); and RGWC + high plantain rate (PLH)), five replicate plots, and repeated in two sequential grazing periods. Forty-five lactating Friesian × Jersey cows were randomly assigned into three groups of 15 animals each to graze over six days in adaptation paddocks and three days in experimental plots. Urine flow sensors were used to measure urine volume and urinary frequency, while spot urine sampling was conducted to determine nitrogen (N) concentration in cow urine. The results showed that including 25% plantain in the diet (PLH) increased daily urine volume by 44% and the daily number of urinations by 28%, compared to grazing the RGWC pasture. In addition, N concentration in cow urine was decreased by 18 and 29% when the diet contained 18% and 25% plantain, respectively. In conclusion, under a typical dairy farm practice, incorporating plantain into the RGWC pasture with the proportion of 25% plantain in the diet can increase the number of urine patches and reduce the concentration of N in the urine, thereby providing the opportunity to decrease N leaching from pastoral systems.

## 1. Introduction 

Nitrogen (N) excreted in cow urine is the main source of N losses from pastoral dairy systems worldwide [1]. Dairy cows grazing traditional swards deposit onto pastoral soils in small localised urine patches, with an N rate between 200 to 2000 kg N/ha on a wetted area between 0.14–0.49 m^2^ per urination [2]. Ideally, the N in soils is used by microbes and retained in organic form or used by pastures in mineral form. The average N load in urine patches usually exceeds the N requirement of pastures, resulting in the risk of nitrate leaching and nitrous oxide emissions [3,4]. On average, only 41% of urinary N (UN) can be taken up by pastures, while 20% of UN could be lost as nitrate leaching from pastoral systems [2]. Regulations have been developed to limit the amount of N loss from intensive dairy farms into the environment to improve water quality [5,6]. Therefore, dairy farmers are required to incorporate strategies to manage their farms and meet regulations on N losses. Mitigation strategies often target reducing the concentration of N in the urine of cows to reduce the amount of UN excreted into soils by dairy cows [4]. 

Plantain (*Plantago lanceolata* L.) has been increasingly used in pastoral systems to overcome feed deficits of conventional perennial ryegrass (*Lolium perenne* L.) and white clover (*Trifolium repens* L.) (RGWC) pastures and improve the milk production of dairy cows, especially during the dry summer, due to its tolerance to drought and heat [7]. In recent years, plantain has emerged as a forage pasture with attributes that reduce N losses from pastoral dairy systems [8]. Particularly, the risk of UN being leached is high in late summer and autumn when herbage growth rate is low, causing more nitrate N being accumulated in soils [9]. Research has consistently confirmed the role of plantain in reducing N concentration in cow urine [9,10] and altering the urination behaviours of dairy cows [11,12]. In addition, these changes are influenced by plantain proportion in the diet, in that a certain composition of plantain is required to have a measurable effect [12,13]. In practice, farmers are challenged to achieve sufficient quantities of plantain in the pasture. Furthermore, in autumn, when the effect of plantain on reducing N loss is important, the content of plantain in the diet is further diluted by use of supplements to offset the decline in pasture growth.

There is limited research on the impact of different proportions of plantain in grazed diets. Most existing studies have either been conducted with feeding cut pastures indoors [12] or grazing swards without supplements [11,14]. Nkomboni, et al. [13] showed that increasing plantain content from 20–70% reduced urinary N concentration in spot samples of grazing cows in late lactation. However, because the nitrate leaching risk is driven by the N load from urine, information is required about both the N concentration and volume of urine [15,16]. The effect of different proportions of plantain on the N load risk has not been elucidated in previous research. In practice, achieving consistently high proportions of plantain (>20%) in the diet can be challenging, particularly when feeding with supplements. Previous research has shown that less than 15% plantain in the diet is unlikely to affect urination behaviour [12,13]. The study by Minnee et al. [12] was carried out under cut-and-carry conditions, and Nkomboni et al.’s [13] study measured urine N% but not urine volume. More information is required regarding the effect of low-to-moderate plantain proportion on the urination behaviour of grazing dairy cows.

Farmer adoption of alternative feeds for mitigating N loss risk requires information on both environmental and production outcomes in order to assess the impact on their farm system and build confidence. The objective of the present study is to determine the impact of RGWC-based pastures containing sizeable percentages of plantain under a typical grazing practice on milk production and the urination behaviour of late-lactation dairy cows. 

## 2. Materials and Methods

The experiment was conducted over two sequential grazing periods, between 23–31 March 2021 and between 5–13 May 2021, at Massey University No. 4 Dairy Farm, Palmerston North, New Zealand (40°23′27″ S 175°36′44″ E), according to the experimental procedure approved by Massey University Animal Ethics Committee (MUAEC #21/09). The study site is on Tokomaru silt loam soil [17] and has a temperate climate. The detailed soil and climate conditions, pasture establishment and management were described in [18].

### 2.1. Experimental Design and Management

The experiment was a completely randomised design with three pasture treatments, five replicate plots (800 m^2^/plot) and two repeated grazing periods (09 days/period). Pasture treatments were perennial ryegrass (*Lolium perenne*, cv. ONE^50^) and white clover (*Trifolium repens* cv. Tribute) (RGWC), RGWC + low proportion of plantain (*Plantago lanceolata*, cv. Agritonic) (PLL) and RGWC + high proportion of plantain (PLH). The swards were established in April 2019 by direct drill and rotationally grazed by dairy cows, with nine grazing events yearly. The sowing rates were designed to have low and high rates of plantain using seed rates where plantain seed was approximately 20% or 35% of the sowing rate and reflected options farmers might use in practice. The sowing rates of plantain, ryegrass, and white clover at sowing were 0, 20, 3 kg in RGWC; 4, 15, 3 kg in PLL; and 7, 10, 3 kg in PLH, respectively. Urea fertiliser (46% N) was applied at 90 kg N/ha/year, with the latest application at 30 kg N/ha on 20 November 2020. The pasture was mown on 13 February 2021 to a consistent height of 6 cm, before the study commenced in March 2021. After each defoliation event in February 2021 and March 2021, the pasture was monitored (using a rising plate meter) to achieve a compressed height of 16–18 cm (approx. 2800 kg dry matter (DM) kg/ha) to start the experimental periods.

Forty-five Jersey Friesian lactating cows were selected from a list of 100 approved cows and assigned to their pasture treatments in each grazing period. Group balance was based on milk yield (19.0 ± 3.4 kg/day in March 2021 and 17.1 ± 3,9 kg/day in May 2021), days in milk (216 ± 13 days in March 2021 and 240 ± 13 days in May 2021), live weight (566 ± 29 kg in March 2021 and 554 ± 43 in May 2021), and somatic cell count (95,000 ± 20,100 in March 2021, and 335,000 ± 61,000 in May 2021) (mean ± SE). Approved cows grazed the same traditional RGWC pastures and were managed under a similar farm practice as in this research before the experimental periods. The data used for group balance were obtained from the milking system three days before each experimental period (milk yield, DIM and live weight) and from a herd test conducted a week before the experiment periods (somatic cell count). In each nine-day grazing period, cows were adapted to their treatment pastures over the first six days by strip grazing together in their mobs of 15 cows (approx. 130 m^2^/cow/day) using temporary fencing materials. Then, each group of 15 cows was subdivided into 5 replicated mobs of 3 cows/mob and randomly allocated for grazing over 3 days in experimental plots (3 cows per each 800 m^2^ plot). At the end of the first grazing period, cows were returned to the farm, and another 45 cows were selected to use in the second grazing period.

During the experimental periods, cows were managed under a typical farm practice, milked twice daily at 07:00 h and 15:00 h. Pasture and supplement allocation to meet energy requirements resulted in approximately 16 kg DM pasture diet above a target postgrazing residual of 1500 kg DM/ha, and five kg DM of supplement. The pasture area allocated for each day was estimated by a rising plate meter and adjusted every half-day by visually assessing the postgrazing residual. For instance, the area was increased when the residual was below the target and vice versa. Supplements were fed in a trough on a concrete feed pad once daily after morning milking by feeding all experimental cows at once over two hours. Supplement composition and group intake were not determined during the study, as all cows were fed together in a single group, though utilisation was over 90%. The supplement was provided with 3 kg maize silage, 1 kg corn gluten pellets and 1 kg pasture baleage. The chemical composition of the supplement was not measured; however, typical maize silage, corn gluten pellets and pasture baleage in New Zealand have respective DM contents of 33, 89 and 38%; metabolisable energy (ME) of 10.3, 12.7 and 9 MJ/kg DM; and crude protein (CP) of 8, 23 and 15% [19]. After finishing the supplemented feeds, the cows were sent to graze their pastures. Cows spent approximately 18 h grazing in the paddocks and 6 h milking and eating supplements. Fresh water was available ad libitum from troughs in each treatment plot and at the feed pad. The following minerals were added (per cow/day) to drinking water through an in-line dispenser: 40 g coarse salt (containing grade 22 NaCl), 40 g MgO (54% Mg), 05 g AquaTrace (10 mg Co, 150 mg Cu, 04 mg I, 04 mg Se and 60 mg Zn).

### 2.2. Herbage Measurements

Pastures were measured for individual plots to determine the herbage DM intake, botanical composition, and nutritive value. Herbage DM mass was measured by randomly harvesting three quadrat cuts to ground level (0.1 m^2^) a day before cows started grazing in the plots for pregrazing and were repeated a day after the grazing for postgrazing. The samples were cleaned of soil and faecal debris and oven-dried at 75 °C until a constant weight was achieved. Herbage DM intake was estimated for each group of cows grazing in individual plots via the following equation: DM intake (kg DM/cow/day) = (pregrazing mass (kg DM) − postgrazing mass (kg DM)) ÷ (no. of cows × no. of grazing days). Trough water intake was calculated with a similar procedure using the data recorded by a flow meter (Gardena Water Smart Flow Meter) attached to a water trough in each plot. Nutritive value and botanical composition of the pastures were measured with a hand-plucked sample to the grazing height (approx. seven cm) taken with 15–20 grabs (approx. 400 g fresh weight) by zigzag pattern across each plot at 10:00 h on a day before cows started grazing in the plots. These samples were placed in plastic bags and stored at 4 °C in a cold room to prevent water evaporation before being weighed. Each sample was mixed thoroughly and subsampled into two smaller samples. The first subsample (approx. 100 g fresh weight) was manually separated into plantain leaves, plantain stem and seed head, perennial ryegrass, white clover, weeds, and dead materials, then oven-dried at 75 °C until a constant weight was achieved to calculate the botanical composition of each component. The second subsample was recorded for fresh weight and oven-dried at 60 °C until a constant weight was achieved to determine the DM content of the pastures. Then, the dried samples were ground to pass a 1 mm sieve for chemical analysis. The analyses were conducted by a commercial laboratory (Hill Laboratory, Hamilton, New Zealand), using near-infrared spectroscopy (NIRS) method for organic matter (OM), crude fat (CF), CP (N × 6.25), acid detergent fibre (ADF), neutral detergent fibre (NDF), nonstructural carbohydrate (NSC) (NSC = 100 − (CP + ash + CF + NDF), OM digestibility, ME (calculated from OMD), Na, K, Ca and S (nitric acid/hydrogen peroxide digestion) [20]. In addition, plant bioactive compounds aucubin and acteoside were analysed with high-performance liquid chromatography at Massey University [21].

### 2.3. Animal Measurements

Milk yield was recorded for each cow twice daily during the morning milking at 07:00 h and the afternoon milking at 15:00 h with an automatic system (Waikato Milking System). In addition, a milk sample (approx. 30 mL) was collected from each cow during morning and afternoon milking on day seven of the experimental periods. Milk samples were analysed for total solids (vacuum oven, AOAC 990.19, 990.20), protein content (Dumas method for N, P = 6.25 × N), and milk urea nitrogen (MUN) (Urease Kinetic UV assay). Urine and faecal spot samples were collected from each cow after morning milking at 09:00 h on days seven and eight of the experimental periods. Urine samples (approx. 80 mL each) were taken by vulva stimulation and subsampled into two smaller samples (approx. 30 mL each). The first subsamples were acidified with sulfuric acid 6.0 N to reduce pH to below 4.0 and analysed for N concentration (Dumas method). The second subsamples were analysed for the content of creatinine (Jaffe method), Na, K, and Cl (ion-selective electrode method). Faecal samples were taken by rectal stimulation and recorded for fresh weight. Then, these samples were freeze-dried until a constant weight was achieved to estimate DM content. The dried samples were ground to analyse for N concentration (Dumas method) and nonprotein nitrogen (NPN) (colourimetric method). All milk, urine and faecal samples were stored at −20 °C until analysed. The analyses of these samples were conducted by a certified laboratory (Nutrition Laboratory, Massey University). 

### 2.4. Urine Volume and Urination Frequency

Daily urine volume, urination volume and the number of urinations were measured using Lincoln University PEETER V1.0 urine flow sensors [22]. In each grazing period, 16 cows were selected from the 45 experimental cows to wear urine sensors over two runs of 36 h for a 24 h measurement (8 cows per run). In the first run, eight urine sensors were attached to the vulva of eight cows (as shown in Figure 1) on day six following the morning milking and were removed following the afternoon milking on day seven. In the second run, sensors were attached to eight cows on day seven after the afternoon milking and removed on day nine following the morning milking. After milking, cows were moved to a vet race where the sensors were attached, a process which took about 45 min and allowed cows to become accustomed to wearing the sensor. The number of cows wearing sensors was balanced in each run with 2 or 3 cows per group, resulting in the final numbers of 11 cows in RGWC, 10 cows in PPL and 11 cows in PPH for both grazing periods. Sensor attachment was monitored as cows walked between the paddock and the milking shed and in the milking parlour at each milking. Sensors that became detached and were leaking urine were immediately reattached or recorded for data exclusion.

### 2.5. Statistical Calculations and Analysis

Variables of herbage (DM mass, botanical composition and nutritive value), animal (milk, urine and faecal samples), and sensor (urine volume and the number of urinations) were analysed using PROC mixed procedure of SAS [23] according to the model: Y_ijk_ = μ + T_i_ + T_i_(G_j_) + C_k_ + e_ijk_, where Y_ijk_ = dependent variable; μ = overall mean; T_i_ = fixed effect of pasture treatment I; T_i_(G_j_) = fixed effect of treatment i nested within grazing period j; C_k_ = random effect (plot for herbage variables and apparent intake or cow for animal and sensor variables), and e_ijk_ = residual error. Comparisons of means were analysed using Fisher’s least significant difference test. A significant difference was declared at *p* < 0.05, and a tendency was declared at *p* < 0.10.

The N concentration in the urine was estimated as the mean of spot samples collected from individual cows over two measurement days. Nutritive value traits, botanical composition of the pastures, herbage DM intake and water intake at the paddock (from herbage intake and water troughs) were analysed with five replicate plots per treatment in each grazing period (*n* = 30, five plots × three treatments × two grazing periods). In addition, variables measured with animal spot samples (milk, urine and faeces) were analysed with the replicated individual cows (*n* = 90, 15 cows × three treatments × two grazing periods).

The data on urine volume and urination frequency from the sensors were scanned to either be excluded from or included in the dataset. A sensor was eligible when it recorded data for at least 24 h. The 24 h was determined from the second urination to mitigate the effect of sensor attachment on the first urination. Event data were excluded when the time from the prior urination was less than five minutes, the event duration was less than five seconds, or events had a similar curve and volume as the previous one [11,24]. The data from six sensors were excluded from the dataset because they were recorded for less than 24 h (4) or sensors were noted as leaking urine (2). As a result, 26 cows wearing sensors were eligible for urine volume and urination frequency, with ten in RGWC, eight in PLL, and eight in PLH. There was a total of 651 urinations recorded over 36 h while the cows wore sensors, in which 409 urination events occurred during 24 measurement hours.

## 3. Results

The average daily temperature and rainfall during the regrowth and grazing periods are presented in Table 1. In general, the weather during the regrowth and grazing periods in March 2021 was hotter and drier than that in May 2021. In addition, sunrise and sunset occurred between 07:36 and 19:20 h during the grazing period in March 2021 and 07:22 h and 17:17 h during the grazing period in May 2021. 

### 3.1. Herbage Characteristics

Herbage DM mass, botanical composition, nutritive value, mineral content and bioactive compounds of the pastures over two grazing periods are presented in Table 2. Pre-grazing DM mass was similar between treatments (2788 ± 253 kg DM/ha) (*p* > 0.10). However, the residual herbage of PLH and PLM tended to be lower than in RGWC (*p* < 0.10). Cows were allocated an estimated 30 kg DM to ground level from the pastures in an area of 130 m^2^ daily during the experimental periods.

The proportion of plantain was 243 g/kg DM in PLL and 342 g/kg DM in PLH (*p* < 0.05), in which reproductive stem and seed head accounted for less than 10 g/kg DM in all treatments. The proportion of perennial ryegrass and dead materials in PLL and PLH was lower than in RGWC (*p* < 0.05). There was no difference in white clover content between treatments (*p* > 0.10). Compared to RGWC, PLL and PLH had a lower composition of DM (%), OM, CF and NDF but contained a higher composition of NSC, K, Na, Ca and S (*p* < 0.05). The higher plantain content in PLH resulted in this treatment having a lower DM%, but higher K, Ca, aucubin and acteoside than PLL (*p* < 0.05).

### 3.2. Milk Production 

The apparent herbage intake of cows was similar between treatments, averaging 15.0 ± 2.5 kg DM/cow/day (*p* > 0.10). Together with supplemented feed, the total apparent intake was 19.5 ± 2.5 kg DM/cow/day (*p* > 0.10). Plantain intake was 3.5 kg DM in PLL and 4.9 kg DM in PLH, accounting for 18 ± 2.1 and 25 ± 2.6% of the total diet (*p* < 0.01).

There was no difference in daily milk yield, total milk solids (content and yield) and milk protein (content and yield) between cows grazing RGWC, PLL and PLH (Table 3, *p* > 0.10). Over two grazing periods, cows produced an average of 17.6 ± 2.9 kg milk, 2.54 ± 0.40 kg total milk solids and 0.69 ± 0.13 kg milk protein. Milk urea nitrogen was the only milk component affected by pasture treatment, with MUN in PLH and PLL being 22% and 12% lower than in RGWC, respectively (*p* < 0.05).

### 3.3. Urine and Faecal Characteristics

Urine volume, urination frequency, urine and faecal characteristics and water intake of dairy cows grazing different pastures are presented in Table 4. Cows grazing low (PLL) and high (PLH) plantain-based pastures urinated 28 and 44% more total daily urine volume than RGWC (*p* < 0.05). The increased volume was excreted through an increased number of events as the average volume per urination did not differ between treatments. The concentration of N in the urine of cows grazing PLH and PLL was 29% and 18% lower than when grazing RGWC (*p* < 0.05). Urine creatinine and faecal N were not different between treatments (*p* > 0.10). Regarding mineral content in the urine, cows grazing plantain-based pastures (PLL and PLH) had a higher Cl, similar K and lower Na than when grazing RGWC (*p* < 0.05).

Water intake from the herbage diet was greater in cows grazing PLH and PLL than those grazing RGWC. Cows partially compensated by consuming less trough water when more water was consumed in the herbage (*p* < 0.05). Compared to RGWC, grazing PPL and PLH increased herbage water intake by 22 and 29 L/cow/day and decreased water intake from drinking troughs by 07 and 10 L/cow/day, respectively. As a result, the water intake from herbage and water troughs at the paddock of cows grazing PLL and PLH was 15 and 19 L more than that of cows grazing RGWC (*p* < 0.05). Regarding water output, cows grazing PLH and PLL excreted more water in urine (*p* < 0.05) but had a lower water content in faeces (*p* < 0.05). 

## 4. Discussion

The results of the present study indicate that under the farm practice of offering both grazed pastures and supplemented feeds, grazing plantain-based pastures increases volume and urination frequency while reducing N concentration in the urine with an influence of the percentage of plantain in the diet. In addition, increasing plantain to up to 25% of a supplemented pasture diet had no effect on milk yield, milk solids and milk protein.

Milk yield of dairy cows fed with the diet containing plantain has been reported as either maintained or increased, depending on the stage of lactation and the difference in DM intake (DMI) and energy intake between experimental and control diets [10,13,25]. The higher DMI and ME intake can improve energy and protein balance to increase milk production when the diets have a similar protein intake [26]. The current results are consistent with previous studies, where DMI and ME content were similar between plantain-based pastures and RGWC pastures, resulting in similar milk yield and milk solids of late-lactation dairy cows. Furthermore, several studies have reported an increased milk protein yield of dairy cows by including plantain in their diet. This is due to the higher ME and undegraded N content in plantain-based pastures than in RGWC pastures that is suggested to improve the utilisation of N by partitioning more N to milk, rather than to urine [12,27]. In the present study, there was no difference in ME content and herbage N intake between plantain-based pastures and RGWC pastures. Although undegraded N intake was not measured, it may not have been much difference between pasture treatments when the diet contained 25% plantain or less. Consequently, grazing plantain-based pastures did not affect milk protein yield in this study.

In the current research, dairy cows grazing pastures containing plantain had increased urine volume, which is most likely driven by the difference in DM content between plantain-based pastures and RGWC pastures. Specifically, the higher water content in plantain compared with RGWC can increase the water intake of dairy cows [11,12], resulting in more water excreted to urine if other water outputs are not affected [24,28]. In the present study, including 18 and 25% dietary plantain resulted in an increase of 22 and 29 L/cow/day in herbage water intake. The higher herbage water intake reduced drinking water from the troughs but increased the total apparent water intake at the paddock by 16 and 20 L/cow/day. At the same time, the water excreted to milk was similar between treatments, while the water excreted to faeces of the cows grazing plantain-based pastures might be lower than when grazing RGWC due to the lower faecal DM content. Therefore, the excessive water input is likely excreted into the urine, increasing urine output by 9 and 18 L/cow/day. A similar effect of water intake on urine output was observed when comparing the data between two grazing periods. The lower herbage DM content in May 2021 compared with that in March 2021 resulted in 34 L more in the water intake from herbage and water troughs at the paddock and 27 L more in the urine volume of cows. The available facility only allowed us to measure water intake from herbage and water troughs from the experimental plots, but not the water intake during supplementation at the barn. Therefore, further studies with more urine sensors and the entire water balance are required to estimate the relationship between water intake and urine volume and urination frequency of dairy cows that are known to affect N losses from pastoral systems [29,30].

The increase in urine volume in the farm condition of this study was observed when cows grazed a diet containing 25% plantain or consumed 4.9 kg DM plantain per day. This effective plantain composition is lower than in previous studies [12,14], which reported that 30% or lower dietary plantain does not affect urine volume. This difference might be due to variations in daily urine volume that greatly vary between animals and sampling days [31,32]. The high number of cows used in the present study (8–10 for urine volume and 15 for UN concentration) allowed us to demonstrate the significant effect of 25% dietary plantain when measured with urine sensors. However, in the present study, urinary creatinine from spot samples was not different between cows grazing plantain-based pastures and RGWC pastures. Differently, in a longer-term measurement period at Massey University using more cows, urine creatinine of cows grazing the same plantain-based pastures (PLL and PLH) was lower than that in RGWC (unpublished data). These results suggest that the creatinine method may underestimate the effect of plantain-based pastures on urine volume when the diet contains a low percentage of plantain, the sample size is small, and spot samples were collected once daily. The diurnal changes of urine creatine require a strong effect (by a high proportion of plantain), or more sampling points or possibly more replicate cows to have a representative estimation of urine volume from urine creatinine [33]. In practice, models can be developed to estimate the impact of plantain-based pastures on urination parameters with a lower percentage of plantain in the diet when sufficient input data are achieved [15,34]. 

Although the current study measured urination behaviour for a relatively short period, the present study agrees with previous research regarding the effect of plantain on increasing urine volume [11,12]. Similarly, our results align with previous research that greater urine production is associated with increased urination frequency and reduced N concentration in the urine rather than the volume of each urination when plantain is included in the diet. The increase of 20 and 44% in urine volume and the reduction of 18 and 29% in UN concentration suggested that UN excretion was probably similar between all treatments. This result aligns with previous studies that show that at least 30% plantain in the diet is required to have a measurable effect on reducing UN excretion by more than 10% [9,12]. Although the mechanisms for reducing UN excretion are not fully understood, the lower UN excretion from plantain-fed cows with similar N intake as control animals is generally associated with greater partitioning of N to dung and milk [12,31]. This is through a combination of improved rumen synchrony of energy and protein (improvement in milk N) and lower protein degradability (greater faecal N). The high NSC content and the presence of the bioactive compounds acteoside and aucubin in plantain have been reported as factors that influence rumen fermentation and microbial N production to improve N utilisation [21,31].

When the total amount of UN excretion is maintained, the dilution of UN is, in part, a key driver from plantain-based pastures to reduce nitrate leaching at the paddock level. The implications of these results can be realised when considering the urine patch area and N load. Using the equation in Romera, et al. [15], as shown in Appendix A, the area affected by urine patches from a single animal (m^2^) can be estimated. Based on the volumes presented in Table 4, the area affected by urine is 4.1, 4.9 and 5.9 m^2^/cow/day for RGWC, PLL and PLH, respectively. Assuming that the lower urinary N concentration of plantain-fed cows was consistent across all urinations, then the N loads under the affected area were 553, 470 and 403 kg N/ha equivalent (estimated using Appendix A). Previous lysimeter research with varying N loads from 0 to 800 kg N/ha have demonstrated a positive relationship between N load and N leaching [15,32], in addition to effects of plantain itself inhibiting nitrification in the soil [35,36]. While incorporating plantain into RGWC pastures can maintain milk production (as in this study) and pasture production [18], its effect on increasing the number of urine patches and reducing UN concentration could provide a successful low-cost option to reduce N leaching from pastoral systems. The increased urine volume and frequency can distribute a similar UN into a larger area with a lower N load. As a result, pastures have the opportunity to uptake more N from urine patches, resulting in a lower risk of N losses from pastoral systems [4,37]. Using the APSIM model, Ledgard, et al. [32] indicated that the increase of 40% in UN concentration with a similar UN excretion (by increasing urine volume and urination frequency) led to a reduction in nitrate leaching by 50% from urine patches and 22% from grazing paddocks.

## 5. Conclusions

In a typical grazing practice, grazing pastures containing 25% plantain in the diet increased urine volume by 44% and the number of urine patches by 28%. In addition, pastures containing 18 and 25% dietary plantain reduced N concentration in cow urine by 18 and 29%. The ability of pastures containing less than 25% plantain to reduce nitrate leaching from pastoral systems is likely associated with increasing urine volume and urination frequency, rather than decreasing UN excretion. Further studies are necessary to investigate the effect of plantain-based pastures on nitrate leaching at the paddock scale and the effective mechanisms, before adopting this technology on a wider scale.

## Figures and Tables

**Figure 1 animals-13-00528-f001:**
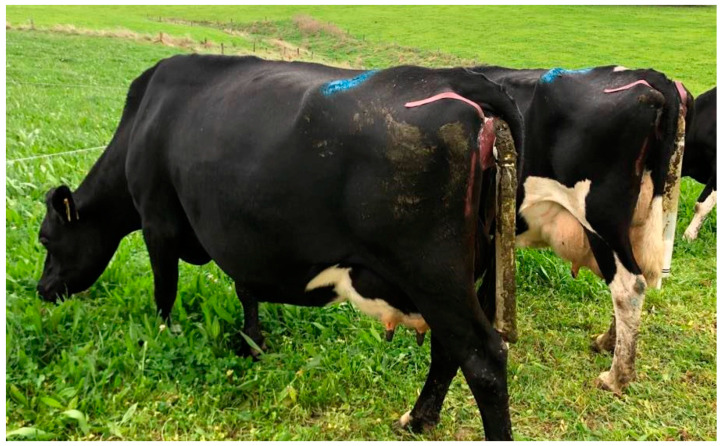
PEETER V1.0 urine sensors attached to cows during the experiment.

**Table 1 animals-13-00528-t001:** Average temperature and the total rainfall during the regrowth and grazing periods.

Variable	Average Temperature (°C)	Total Rainfall(mm)
Regrowth period before March grazing	17.1	1.9
Grazing period in March grazing	17.1	1.2
Regrowth period before May grazing	14.4	2.9
Grazing period in May grazing	15.3	1.6

Note: climate data were achieved from a climate station 500 m from the experimental site.

**Table 2 animals-13-00528-t002:** Herbage mass, botanical composition and nutritive value of pastures containing ryegrass and white clover (RGWC) or with a low (PLL) or high (PLH) proportion of plantain.

Variable ^1^	RGWC	PLL	PLH	SEM	*p*-Value
*Herbage mass*					
Pregrazing mass (kg DM/ha)	2815	2784	2745	62	0.725
Postgrazing mass (kg DM/ha)	1551	1425	1358	61	0.096
Allocation area (m^2^/cow/d)	130	130	130	-	-
DM allocation (kg/cow/d)	30.0	29.7	29.3	0.66	0.701
*Botanical composition*					
Plantain (g/kg DM)	-	243 ^a^	342 ^b^	17	<0.001
Perennial ryegrass (g/kg DM)	611 ^a^	380 ^b^	305 ^c^	21	<0.001
White clover (g/kg DM)	236	266	249	16	0.409
Dead material (g/kg DM)	145 ^a^	107 ^b^	103 ^b^	7.9	0.001
*Nutritive value*					
Dry matter (%)	20.4 ^a^	17.6 ^b^	16.3 ^c^	0.25	<0.001
OM (%)	86.8	85.6	84.9	0.33	0.004
Crude fat (g/kg DM)	35 ^a^	29 ^b^	29 ^b^	0.7	<0.001
Crude protein (g/kg DM)	231	232	234	5.1	0.889
ADF (g/kg DM)	268	255	254	4.3	0.060
NDF (g/kg DM)	484 ^a^	441 ^b^	424 ^b^	8.0	<0.001
NSC (g/kg DM)	124 ^a^	154 ^b^	162 ^b^	8.0	0.008
OM digestibility (g/kg DM)	719	718	722	7.8	0.919
ME (MJ/kg DM)	10.0	9.8	9.8	0.12	0.418
Na (g/kg DM)	4.9 ^a^	5.9 ^b^	6.5 ^b^	0.31	0.004
K (g/kg DM)	26 ^a^	28 ^ab^	30 ^c^	0.90	0.011
Ca (g/kg DM)	8.4 ^a^	14.0 ^b^	15.8 ^c^	0.52	<0.001
S (g/kg DM)	13 ^a^	17 ^b^	19 ^b^	0.7	<0.001
Aucubin (g/kg DM)	0.18 ^a^	2.68 ^b^	4.58 ^c^	0.34	<0.001
Acteoside (g/kg DM)	0.29 ^a^	6.77 ^b^	10.86 ^c^	0.71	<0.001

^a,b,c^ Mean values in the same row with different superscripts differ (*p* < 0.05). ^1^ ADF = acid detergent fibre; DM = dry matter; NDF neutral detergent fibre; NSC = nonstructural carbohydrates; ME = metabolisable energy; OM = organic matters; SEM = standard error of the mean.

**Table 3 animals-13-00528-t003:** Herbage intake, milk yield and total milk solids, milk protein and milk urea nitrogen of cows grazing RGWC, PLL and PLH.

Variable	RGWC	PLL	PLH	SEM	*p*-Value
Herbage intake (kg DM/cow/day)	14.1	15.2	15.4	0.65	0.329
Total apparent intake (kg DM/cow/day) ^1^	18.7	19.6	19.8	0.77	0.546
Plantain intake (kg DM/cow/day)	0.0 ^a^	3.5 ^b^	4.9 ^c^	0.32	<0.001
Plantain content of the diet (%)	0.0 ^a^	17.7 ^b^	24.6 ^c^	1.45	<0.001
Milk yield (kg/cow/day)	18.1	17.6	17.4	0.52	0.577
Milk solids (g/100 g milk)	14.4	14.8	14.3	0.21	0.231
Milk solids (kg/cow/day) ^2^	2.63	2.61	2.47	0.07	0.178
Milk protein (g/100 g milk)	3.9	4.1	3.9	0.07	0.135
Milk protein (kg/cow/day)	0.71	0.71	0.66	0.02	0.104
Milk urea nitrogen (mg/dL)	13.8 ^a^	12.1 ^b^	10.8 ^c^	0.25	<0.001

^a,b,c^ Mean values in the same row with different superscripts differ (*p* < 0.05), ^1^ assuming cows utilised 90% of provided supplemented feeds; ^2^ milk solids include fat, protein, lactose and ash.

**Table 4 animals-13-00528-t004:** Urine volume, urination frequency, urine and faecal characteristics and water intake of dairy cows grazing RGWC, PLL and PLH.

	RGWC	PLL	PLH	SEM	*p*-Value
Apparent herbage N intake	507	539	560	26.8	0.392
*Urine volume and urination pattern*					
Urine volume (L/cow/day) ^1^	40.8 ^a^	49.1 ^ab^	58.9 ^b^	3.66	0.008
Urination (urinations/cow/day) ^1^	13.3	14.8	17.0	1.05	0.064
Urination volume (L/urination) ^1^	3.0	3.5	3.5	0.22	0.187
Urinary N (g/kg) ^2^	5.5 ^a^	4.5 ^b^	3.9 ^b^	0.23	<0.001
Urine creatinine (mmol/L) ^2^	2406	2140	2270	106	0.199
Urine urea (mmol/L) ^2^	242 ^a^	197 ^b^	172 ^b^	7.0	<0.001
Urine NH_3_ (mmol/L) ^2^	2.5	2.8	2.7	0.15	0.427
Urine Na (mmol/L) ^2^	88 ^a^	75 ^b^	72 ^b^	3.84	0.008
Urine K (mmol/L) ^2^	169	176	176	4.6	0.457
Urine Cl (mmol/L) ^2^	121 ^a^	171 ^b^	181 ^b^	6.8	<0.001
*Faecal characteristic*					
Faecal DM content (%) ^2^	11.3 ^a^	12.6 ^b^	13.6 ^c^	0.34	<0.001
Faecal N (g/kg DM) ^2^	27.7	28.1	27.4	0.38	0.485
*Water intake*					
Herbage water intake (L/cow/day) ^3^	57 ^a^	79 ^b^	86 ^b^	3.6	<0.001
Trough water intake (L/cow/day) ^3^	23 ^a^	16 ^b^	13 ^b^	1.5	<0.001
Total apparent water intake (L/cow/d) ^4^	85 ^a^	101 ^b^	105 ^b^	4.0	0.015

^a,b,c^ Mean values in the same row with different superscripts differ (*p* < 0.05). ^1^ calculated with the data from cows that wore urine sensors (n = 26) with 10 in RGWC, 8 in PLL and 8 in PLH; ^2^ calculated with the data from all cows (n = 90); ^3^ calculated with the data from pasture plots (n = 30); ^4^ assuming cows had similar water from supplements of 5.5 L/cow/day, estimated with the supplement DM% of 45% and the utilisation of 90%.

## Data Availability

The data presented in this study are available on request from the corresponding author.

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
