# Peer review of "Dairy Cows Grazing Plantain-Based Pastures Have Increased Urine Patches and Reduced Urine N Concentration That Potentially Decreases N Leaching from a Pastoral System"

_animals, 2023, doi:10.3390/ani13030528_

Round 1
Reviewer 1 Report
This paper is focused on determining the effect of grazing plantain-based pastures 25 on urine volume, urination frequency, and urinary nitrogen (UN) concentration of dairy cows under 26 a typical pastoral dairy practice. Here are my comments:
Major comments:
My concern with this paper is the experimental duration. Data were only collected for 3 days in both grazing periods. Why do authors think this is long enough to get strong measurements?
Cows were used in the statistical model. Since treatments were not directly applied to cows, I don’t think this is correct way of analyzing data. Detailed comments in later part of the review.
Urine samples were only collected once each day and it is not representative and will not represent 24-h values. Since this was the most important parameter of this study, I wonder why more samples were not collected.
Simple summary: I believe that the simple summary should be more simply and straight forward written in order to reach a large audience. There is no need to describe the results within each treatment, neither describe how the samples were collected.
L29: Please add rate of plantain inclusion in PLL and PLH treatments.
L33: If N is not an accepted abbreviation, please define at first use.
L47-49: Please rephrase.
L50: Is there any quantification on how much N pasture can take. If there is, please specify it here.
L62: Explain why risk of urinary N loss is high in autumn.
L81-83: Please specify plantain percentages used in the study. Also, relate the percentage used in this study with the proportion used in previous studies and why authors decided to use 18 and 25%? Please add hypothesis.
L96: I understand that pasture management was provided in previous paper but adding details here on the seeding rate and fertilizer management will be useful for readers.
L114: 5 adaptation days + 3 days grazing experimental plots = 8-day grazing. Why 9-day grazing period is mentioned?
L123: How was supplement provided? To all 45 cows together or in group of 3 cows?
L128: I don’t understand how cows are fed in a single group. There are 15 groups of cows (3 cows per group) and 5 groups in each treatment pasture. Measuring intake (pasture disappearance) is critical component of this project when objectives are to measure urinary N.
Table 1. I don’t see the value of this table if these numbers are not from the supplement that was fed to cows. If these are the actual values, please add details on when the supplement was sampled, frequency of sample collection, and how ME was evaluated?
L140: I recommend splitting the measurements in subtopics. For example: Forage mass and botanical comp.; DM and water intake; milk production…
L176: Is just one sample of urine on d 7 and 8 representative of diurnal variability? Same goes for fecal spot samples.
L205: What was the experimental unit? Since treatments were not given to cows, I don’t think cows can be used in the model and regardless of the production parameters collected per cow, it should be presented per grazing plot. Similar analysis should be conducted for urine and fecal parameters.
Table 2. Please add units. Is this data available online from the climate station? If it is, please provide this link.
L241: is this the Herbage mass of each plot or total of every treatment? I suggest these numbers should be for each replicate plot and add measure of variability between plots.
Table 3. The herbage intake was calculated for each plot or for the entire grazing treatment.
Table 3: Please explain if MUN values are in the normal range.
Table 4. I suggest adding number of cows / treatment with urine sensors right below urine volume and urination pattern.
Discussion: What about the bioactive compounds? Why were they measured? How does it help to explain the results?
L306: No hypothesis was provided.
L359: Either this or it may also depend on how representative urine samples are collected in this study. Regardless, why authors believe that feeding plantain diets will underestimate urine volume when creatinine is used as marker.
L376: If greater than 30% plantain proportions are effective, then why authors chose 18 and 25% of plantain proportions?
L378: What dictates N partitioning and why 30% of plantain is effective? Which NSC component is prevalent in plantain?
L382: Explain this equation in Materials and Methods.
L384-386: These results on area affected by urine patches should also be shown in tables in the "Results" section
L392: This is the most important point for this manuscript and is not properly explained. I am still not sure why nitrate leaching will go down when the total urine N applied is similar on the grazing area.
L394: Conclusion is basically summary of your results. I suggest to focus this section on the implications and guidelines on the proportion of plantain in the grazing areas for lactating dairy cows.
Reviewer 2 Report
The abbreviations should be improved.

Author Response
We responded to reviewer 2's comments directly in the PDF file. Please see the attachment.

Round 2
Reviewer 1 Report
I have gone through the revisions and happy to report that authors have addressed my concerns. I have no more comments.